# Emotional and Behavioural Adjustment in Children and Adolescents with Short Stature vs. Their Normal-Statured Peers

**DOI:** 10.3390/jcm14020538

**Published:** 2025-01-16

**Authors:** Anna Guerrini Usubini, Nicoletta Marazzi, Laura Abbruzzese, Gabriella Tringali, Roberta De Micheli, Gianluca Castelnuovo, Alessandro Sartorio

**Affiliations:** 1Experimental Laboratory for Auxo-Endocrinological Research, Istituto Auxologico Italiano, Istituto di Ricovero e Cura a Carattere Scientifico (IRCCS), 28824 Piancavallo-Verbania, Italy; n.marazzi@auxologico.it (N.M.); g.tringali@auxologico.it (G.T.); r.demicheli@auxologico.it (R.D.M.); sartorio@auxologico.it (A.S.); 2Division of Auxology, Istituto Auxologico Italiano, Istituto di Ricovero e Cura a Carattere Scientifico (IRCCS), 28824 Piancavallo-Verbania, Italy; l.abbruzzese@auxologico.it; 3Psychology Research Laboratory, Istituto Auxologico Italiano, Istituto di Ricovero e Cura a Carattere Scientifico (IRCCS), 28824 Piancavallo-Verbania, Italy; gianluca.castelnuovo@auxologico.it; 4Department of Psychology, Catholic University of Milan, 20123 Milan, Italy

**Keywords:** short stature, psychological adjustment, emotional and behavioural problems, children, adolescents

## Abstract

**Background/Objectives**: the aim of the current cross-sectional study is to explore and compare the emotional and behavioural conditions of children and adolescents with short stature (i.e., familial short stature and constitutional delay of growth), idiopathic growth hormone deficiency (GHD), and normal height. **Methods**: Twenty-nine participants (15 males, 14 females; mean age ± standard deviation (SD) = 11.2 ± 1.96 years) with short stature (height standard deviation score (SDS): −2.10 ± 0.57), 10 age-matched participants (4 males, 6 females; mean age ± SD = 10.9 ± 2.35 years) with growth hormone deficiency (GHD; height SDS: −2.44 ± 0.29), and 36 age-matched participants (19 males, 17 females; mean age ± SD = 11.3 ± 1.93 years) with normal stature (height SDS: 0.56 ± 0.78) were admitted to this study. Psychological distress was evaluated using the Depression Anxiety and Stress Scale (DASS-21), and emotional and behavioural problems using the Child Behaviour Checklist for Children (CBCL) and the Strengths and Difficulties Questionnaire (SDQ). **Results**: Participants with GHD exhibited higher levels of the “withdrawn/depressed subscale” score of CBCL (H = 7.794, df = 2, *p* = 0.020), compared to their peers with normal height, while no significant differences were observed between participants with short stature and those with normal stature. Normal-statured participants reported higher levels of the “conduct problems” subscale score (H = 10.421, df = 2, *p* = 0.005) and the “rule-breaking behaviour” subscale score of CBCL (H = 10.358, df = 2, *p* = 0.006) compared to both short-statured groups. No significant differences among the three subgroups were found in the DASS-21 and SDQ scores questionnaires, suggesting the lack of short stature-dependent effects on psychological distress and emotional and behavioural adjustment. **Conclusions**: Although participants with GHD exhibited higher levels of the “withdrawn/depressed subscale” score of CBCL (H = 7.794, df = 2, *p* = 0.020) compared to their peers with normal height, participants with short stature did not experience the same problems. Although short-statured participants had lower levels of “conduct problems” and “rule-breaking behaviours” scores than their normal-statured peers, the psychological distress and emotional and behavioural adjustment were not affected by short stature, being comparable to those recorded in normal-statured participants.

## 1. Introduction

Short stature in children and adolescents is frequently a cause of concern for medical, psychological, and social implications [1]. Although the physical implications represent the primary focus of medical evaluation, the psychological and social effects of short stature on children and adolescents are equally critical. Several studies reported that children with short stature may face unique emotional and behavioural challenges, including bullying, social isolation, and perceived stigma, leading to internalizing behaviours such as anxiety and depression [1,2]. Other authors suggested that social attitudes toward height, including overprotection by parents or lower expectations from teachers, can exacerbate these challenges, shaping a child’s self-perception and social confidence [2].

It is noteworthy that not all short-statured children experience the same psychological difficulties. Several studies reported that their social, emotional, and behavioural outcomes often align with the normative range, suggesting that many children adapt well despite their short stature [3,4]. In this regard, Quitmann et al. [2] reported that social support, resilience, and adaptive coping strategies could play protective roles in mitigating negative outcomes. For instance, children who perceive strong support from peers and family members tend to report better self-esteem and quality of life, irrespective of their height [5]. These contradictory results impose the need to examine the interplays between clinical factors and psychosocial outcomes. In addition, despite substantial research on the psychosocial implications of GHD [6,7], relatively few studies have compared the emotional and behavioural profiles of children with short stature due to different etiologies and even fewer have directly compared these profiles with those of children of normal height. Such comparative analyses are essential to clarify the psychological conditions of children with different forms of short stature in comparison with those of normal height and to disentangle the specific psychological impact of a non-GH-dependent short stature (i.e., familial short stature, FSS or constitutional growth delay, CGD) versus a short stature due to a GH deficiency (GHD).

The aim of the current cross-sectional study is to explore and compare the emotional and behavioural conditions of age-matched children and adolescents with non-GH-dependent short stature (FSS and CGD), idiopathic GHD, and normal height.

## 2. Materials and Methods

### 2.1. Participants and Procedures

The study population of short-statured patients, recruited at the Research Center for Growth Disorders, Istituto Auxologico Italiano, IRCCS, Milan, Italy, was composed of the following groups:

Group 1: 29 participants (15 males, 14 females; mean age ± SD: 11.2 ± 1.96 years; mean height SDS: −2.10 ± 0.57), according to the Italian reference growth charts for age and sex [8], with FSS or CGD. Participants with FSS (10 males, 10 females; mean age ± SD: 11.0 ± 1.9 years; mean height SDS: −2.2 ± 0.6) were characterised by a short stature observed in other family members (not necessarily the parents), height within the expected range for the parental target height, proportionate physical appearance without significant clinical signs, normal timing of pubertal development, and bone age corresponding to the chronological age. Participants with CGD (5 males, 4 females; mean age ± SD: 11.6 ± 2.1 years; mean height SDS: −2.0 ± 0.5) were characterised by a negative familiarity for short stature (i.e., “short for their parents”), proportionate physical appearance without significant clinical signs, absence of any systemic, endocrine, nutritional, or chromosomal abnormality, and delayed bone age.

A normal GH responsiveness during at least one GH-releasing stimulus (i.e., GH peak > 8 ng/mL) excluded the presence of GHD in the 5 short-statured participants with SDS lower than −2.5.

Group 2: 10 participants (4 males, 6 females; mean age ± SD: 10.9 ± 2.35 years, mean height SDS: −2.44 ± 0.29) with idiopathic GH-deficiency (GHD), based on the criteria defined by the Italian Medicine Agency (AIFA) note 39 for this clinical condition. These participants were characterized by short stature, ≤−3 SDS or ≤−2 SDS, with a growth velocity per year ≤ −1.0 SDS for age and sex evaluated over at least 6 months, and GH peak levels < 8 ng/mL during two different pharmacological stimulation tests [9]. A brain magnetic resonance scan was performed to exclude the presence of pathological conditions.

Group 3: 36 participants (19 males, 17 females; mean age ± SD: 11.3 ± 1.93 years; mean height SDS: 0.56 ± 0.78) with normal stature (height SDS ≥ −0.64) were recruited among the sons/daughters of the hospital’s medical, research, and administrative staff and among the sons/daughters of friends and colleagues. Convenience sampling was utilized due to its practicality in accessing readily available populations. Although this approach inherently limits generalizability, efforts were, however, made to mitigate these limitations by targeting participants of the same age, sex, and level of education.

Recruitment was carried out between July 2023 and November 2024.

This study was approved by the Ethical Committee (EC) of Istituto Auxologico Italiano, IRCCS, Milan, Italy (approval number EC: 2023_03_21_03; research code: 01C312; acronym: PSICOSHORT). This research was carried out according to the Declaration of Helsinki and its advancements.

### 2.2. Measures

#### 2.2.1. Anthropometric Data

Weight and height were measured by the internal medical staff. Standing height was determined by a Harpenden Stadiometer (Holtain Limited, Crymych, Dyfed, UK). Weight was measured to the nearest 0.1 kg using an electronic scale (RoWU 150, Wunder Sa.bi., Trezzo sull’Adda, Italy). Body mass index (BMI) was calculated using the formula kg/m^2^.

#### 2.2.2. Emotional and Behavioural Data

Psychological distress was assessed using the Italian version of the Depression Anxiety and Stress Scale (DASS-21) [10,11], a widely used self-report questionnaire designed to assess emotional states of depression, anxiety, and stress. The instrument consists of 21 items, divided equally into three subscales, Depression, Anxiety, and Stress, with seven items each. Respondents rate the frequency or severity of their experiences over the past week on a 4-point Likert scale, ranging from 0 (“Did not apply to me at all”) to 3 (“Applied to me very much or most of the time”). Higher scores on each subscale indicate greater severity of symptoms. Considering the small sample size, the internal consistency of the subscales in our sample was explored using item correlations among items of the same subscale. The results showed that all the items comprising the three subscales of the questionnaires had significant correlations with each other (*p* < 0.05), suggesting a good internal consistency for all the subscales of the DASS-21.

Emotional and behavioural problems were assessed using the Italian versions of the Child Behaviour Checklist for Children [12] and The Strengths and Difficulties Questionnaire (SDQ) [13,14]. The CBCL is a widely used parent-report questionnaire designed to assess behavioural and emotional problems in children and adolescents aged 6 to 18 years. The CBCL 6–18 consists of 113 problem items rated on a 3-point Likert scale, ranging from 0 (“Not true”) to 2 (“Very true or often true”), based on the child’s behaviour over the past 6 months. Items comprise eight empirically derived syndromes (Anxious/Depressed, Withdrawn/Depressed, Somatic Complaints, Social Problems, Thought Problems, Attention Problems, Rule-breaking Behaviour, and Aggressive Behaviour), which are grouped into two general dimensions (internalizing problems and externalizing problems) and six DSM-oriented scales (Affective Problems, Anxiety Problems, Somatic Problems, Attention Deficit/Hyperactivity Problems, Oppositional Defiant Problems, and Conduct Problems).

Correlations among items of the same subscale were computed. The results showed low or no significant correlations between items within the CBCL subscales, probably suggesting poor internal consistency.

The SDQ is a brief screening tool designed to assess emotional and behavioural difficulties as well as prosocial behaviours in children and adolescents aged 4 to 17 years. The questionnaire comprises 25 items divided into five subscales: Emotional Symptoms, Conduct Problems, Hyperactivity/Inattention, Peer Relationship Problems, and Prosocial Behaviour. Each subscale contains five items rated on a 3-point Likert scale (0 = “Not true”, 1 = “Somewhat true”, 2 = “Certainly true”), reflecting the child’s behaviour over the past six months. A Total Difficulties score is calculated by summing the scores from the first four subscales, with higher scores indicating greater difficulties. Correlations among items of the same subscale were computed. The results showed low or no significant correlations between items within the SDQ subscales, probably suggesting poor internal consistency.

#### 2.2.3. Statistical Analysis

Data are presented as frequencies and percentages or means and standard deviations, as appropriate. To analyse differences among the three subgroups, a Kruskal–Wallis test, a non-parametric alternative to the one-way ANOVA suitable for small samples, was used for the data that did not meet the assumptions of normality or homogeneity of variance. This test compares the ranks of the dependent variable across the subgroups to determine if there are significant differences. Post-hoc pairwise comparisons were conducted using the Dunn test with adjustments for multiple comparisons to control for Type I error. The epsilon squared (ε^2^) effect size was computed and interpreted as follows: ε^2^ ≤ 0.01: small effect; ε^2^ ≥ 0.06: medium effect; ε^2^ ≥ 0.14: large effect. The corrections applied included the Holm–Bonferroni method. Correlations between variables were assessed using Kendall’s tau correlation coefficient, a non-parametric measure of association between variables which is suited for small sample sizes. Kendall’s tau evaluates the strength and direction of the monotonic relationship between two variables by comparing the number of concordant and discordant pairs. The coefficient ranges from −1 to +1, where values closer to ±1 indicate stronger associations and values near 0 suggest a lack of correlation.

All analyses were performed using Jamovi (2.5.5.), and a significance level of *p* < 0.05 was set for all tests.

## 3. Results

The study population comprised three subgroups, as reported in Table 1. Age (*p* = 0.867), gender composition (*p* = 0.814), and school attendance (*p* = 0.686) were similar in the three subgroups, while the geographical nation of origin was slightly different due to the local recruitment (Northern Italy) of the normal-statured group. Height SDS was significantly (*p* < 0.001) lower in patients with GHD than in those with normal stature, and it was significantly lower in patients with short stature than those with normal stature (*p* < 0.001). No significant differences were recorded between the height SDS of patients with short stature and those with GHD (*p* = 0.106).

Differences between the three subgroups of our sample in DASS-21, CBCL, and SDQ were explored.

Since no significant differences were detected between participants with FSS and CGD, probably related to the comparable height SDS, the short stature subgroup was considered altogether without a further subdivision.

Significant differences among the three subgroups of our sample were observed in the Withdrawn/Depressed subscale, Conduct Problems subscale, and Rule-breaking subscale scores of the CBCL.

Post hoc comparisons using the Dwass–Steel–Critchlow–Fligner method revealed that participants with idiopathic GHD exhibited higher levels of the “withdrawn/depressed subscale” score of CBCL (H = 7.794, df = 2, *p* = 0.020), compared to their peers with normal height, while no significant differences were observed between participants with short stature (FSS or CGD) and those with normal stature.

Normal-statured participants reported higher levels of the “conduct problems” (H = 10.421, df = 2, *p* = 0.005) and “rule-breaking behaviour” scores of CBCL (H = 10.358, df = 2, *p* = 0.006) compared to the short-statured subgroups (FSS, CGD, and idiopathic GHD).

By contrast, no significant differences among the three subgroups were found in the DASS-21 and SDQ scores questionnaires, suggesting the lack of short stature-dependent effects on psychological distress and emotional and behavioural adjustment.

The means and standard deviations of the DASS-21, SDQ, and CBCL scores are presented in Table 2.

## 4. Discussion

The current cross-sectional study compared participants with short stature (FSS, CGD, and idiopathic GHD) with age- and sex-matched peers of normal stature concerning psychological distress and emotional and behavioural problems. Our results show that participants with idiopathic GHD exhibited higher levels of the Withdrawn/Depressed subscale score of CBCL compared to their peers with normal height (ε^2^ = 0.10 medium effect), while no significant differences were observed between participants with short stature and those with normal stature. In addition, normal-height participants reported higher levels of the “conduct problems” (ε^2^ = 0.14 large effect) and the “rule-breaking behaviour” subscale scores of CBCL (ε^2^ = 0.13 medium effect) compared to both short-statured groups.

By contrast, no significant differences among the three subgroups in DASS-21 and SDQ were found, suggesting no height-related differences existed in psychological distress and emotional and behavioural problems.

The present study appears to support the contradictory nature of the literature regarding the psychological consequences of short stature [15].

On the one hand, numerous studies suggested the presence of adjustment difficulties and poor quality of life in children and adolescents with short stature [16]. Consistently, in our study, participants with idiopathic GHD exhibited higher levels of the “withdrawn/depressed” subscale score of CBCL compared to their peers with normal height. This finding aligns with the existing literature emphasizing the psychosocial challenges faced by this short-statured population. In a previous study, children with idiopathic GHD were reported to have increased emotional difficulties, which were attributed to the dual impact of the condition itself (hormonal deficiency) and the associated physical differences, such as short stature, that can influence self-esteem and social interactions [17]. Differently than patients with idiopathic GHD, participants with non-GH-dependent short stature did not exhibit significantly greater depressive symptoms compared to normal-statured peers. This finding seems to suggest that idiopathic GHD per se, which determines a more severe degree of short stature than that observed in participants with FSS/CGD and is frequently associated with physical problems (reduced muscle mass and force, delayed bone age, childish appearance, etc.), may contribute to the development of emotional difficulties not detected in patients with FSS (who frequently live in a familiar context characterised by other people suffering from short stature, thus limiting the psychological impact of being short). On the other hand, there is also evidence indicating normal development in children with short stature [18]. In agreement with this finding, no significant differences among the three subgroups were found in the DASS-21 and SDQ subscales, suggesting the lack of a stature-dependent effect in psychological distress and emotional and behavioural problems. Although our findings appear to be aligned with the discordant results reported in the literature, it is worth noting that they may also reflect the differences in the tools used to assess similar variables. For example, in the present study, a higher prevalence of depressive symptoms was observed in individuals with idiopathic GHD compared to those with normal height, specifically in the Withdrawn/Depressed subscale of the CBCL, but not in the Depression subscale of the DASS-21. The Withdrawn/Depressed subscale of the CBCL is specifically designed to assess behaviours and emotional states associated with social withdrawal, isolation, and depressive symptoms in children and adolescents. On the other hand, the Depression subscale of the DASS-21 is a self-report measure designed to capture emotional, cognitive, and physical aspects of depression, providing insight into the severity and presence of depressive symptoms, without considering any form of withdrawal, especially in social life. Thus, the subscale of CBCL could be more adapted to detect the social isolation that broadly characterizes participants with idiopathic GHD.

Interestingly, children of normal height showed higher scores in the “conduct problems” and “rule-breaking behaviour” subscales of CBCL compared to participants with FSS-CGD and GHD. These results are similar to those obtained in previous studies showing that children with GHD appeared less aggressive and assertive than their normal-statured peers [19]. Several hypotheses about this finding can be discussed. This result could be influenced by differences in social expectations and adaptations among these subgroups. While children with short stature often experience being treated as younger than they effectively are, potentially fostering more conformist behaviours, those with normal height might not face the same constraints. Moreover, normal-height children may experience broader peer interactions where social pressures contribute to risk-taking behaviours [20]. It is also possible that families of children with short stature adopt more protective parenting styles, which may indirectly shape children’s adherence to rules and reduce externalizing behaviours [18].

The current study presents several limitations to be acknowledged. First, the results of GHD patients were obtained in a relatively small number of participants with this rare disease; thus, we are requesting this to be confirmed in a larger multicentre study population. Additionally, the sample was composed exclusively of participants recruited at a single centre, thus deserving caution in generalising the findings to other populations.

Second, the cross-sectional design prevents any determination of causality between variables. Third, the use of self-report questionnaires, even though consistent with the existing literature, may be affected by response bias. In addition, probably due to the relatively small sample, the internal consistency of the subscales of the questionnaires was low. Lastly, the absence of longitudinal data limits the ability to track changes in the psychological conditions of the participants, mainly of those with GHD during recombinant GH therapy.

Despite these limitations, these findings underscore the importance of considering the psychosocial impacts of short stature. Interventions for children with GHD should address both the physiological aspects of the condition and the psychological burden that may increase their vulnerability to depressive symptoms. Conversely, the higher prevalence of conduct issues among normal-height children suggests a broader need to examine social and familial dynamics that may influence their behaviour.

Future studies should explore longitudinal data to assess whether these patterns persist over time and examine potential mediating factors, such as parental attitudes, peer relationships, and societal perceptions. Additionally, neurobiological research may shed light on the specific pathways linking GHD and depressive symptoms.

## Figures and Tables

**Table 1 jcm-14-00538-t001:** Main characteristics of the study population.

		Group 1 Short Stature *n* = 29	Group 2 Idiopathic GHD *n* = 10	Group 3 Normal Stature *n* = 36
Sex	Males *n* (%)	15 (51%)	4 (40%)	19 (54%)
	Females *n* (%)	14 (49%)	6 (60%)	17 (46%)
Age (yrs)	M (SD)	11.2 (1.96)	10.9 (2.35)	11.3 (1.93)
Height (m)				
Height SDS	M (SD)	−2.10 (0.57)	−2.44 (0.29)	0.56 (0.78)
Weight (kg)	M (SD)	30 (7.87)	28.9 (9.51)	43 (10.3)
Weight SDS	M (SD)	−1.79 (0.73)	−1.85 (0.98)	0.048 (0.66)
BMI (kg/m^2^)	M (SD)	16.8 (2.20)	17.1 (3.41)	18.6 (2.24)
BMI SDS	M (SD)	−0.99 (0.89)	−0.87 (1.30)	−0.32 (0.69)
School attendance	Primary school *n* (%)	13 (42%)	5 (50%)	16 (46%)
	Secondary school *n* (%)	16 (58%)	5 (50%)	20 (54%)
Geographical origin	Northern Italy *n* (%)	20 (72%)	6 (60%)	36 (100%)
	Central Italy	5 (14%)	-	-
	Southern Italy *n* (%)	4 (14%)	4 (40%)	-

**Table 2 jcm-14-00538-t002:** Means and standard deviations in DASS-21, SDQ, and CBCL.

	Group 1 Short Stature *n* = 29 M (SD)	Group 2 Idiopathic GHD *n* = 10 M (SD)	Group 3 Normal Stature *n* = 36 M (SD)	H	*p*
DASS-21					
Depression	2.07 (2.70)	2.50 (2.68)	3.14 (3.89)	0.884	0.643
Anxiety	3.17 (3.36)	3.40 (3.84)	3.00 (3.39)	0.475	0.789
Stress	5.52 (3.93)	6.50 (4.74)	5.64 (4.14)	0.484	0.785
SDQ					
Emotional Symptoms	1.90 (1.70)	2.78 (3.35)	2.25 (2.30)	0.257	0.879
Conduct Problems	2.45 (1.48)	2.67 (1.50)	2.69 (1.53)	0.513	0.774
Hyperactivity/Inattention	4.45 (1.48)	4.11 (1.53)	4.47 (1.38)	0.937	0.626
Peer Problems	4.62 (1.50)	4.56 (0.88)	4.42 (1.00)	0.207	0.902
Prosocial Behaviours	7.97 (1.82)	8.78 (1.20)	8.19 (1.62)	2.064	0.356
Total Difficulties	13.41 (3.90)	14.11 (5.88)	13.83 (4.01)	0.526	0.769
CBCL					
Anxious/Depressed	4.13 (3.43)	3.90 (3.14)	2.58 (2.50)	5.025	0.081
Withdrawn/Depressed	2.03 (2.20)	2.60 (1.65)	0.97 (1.11)	7.794	0.020 *
Somatic Complaints	1.79 (1.82)	2.00 (2.26)	1.89 (1.77)	0.078	0.962
Social Problems	2.48 (2.28)	3.00 (1.89)	2.47 (2.47)	1.460	0.482
Thought Problems	1.76 (2.39)	1.50 (1.43)	1.17 (1.46)	1.231	0.540
Attention Problems	2.90 (2.86)	4.30 (3.53)	3.28 (2.74)	1.622	0.444
Rule-breaking Behaviour	1.24 (1.64)	0.80 (1.03)	2.86 (2.78)	10.358	0.006 *
Aggressive Behaviour	3.79 (2.99)	2.70 (2.21)	2.89 (2.25)	1.517	0.468
Other Problems	2.51 (2.01)	3.11 (2.15)	3.83 (2.94)	2.869	0.238
Affective Problems	2.14 (2.20)	3.10 (1.97)	1.92 (1.87)	3.295	0.192
Anxiety Problems	2.90 (2.62)	2.40 (2.01)	1.53 (1.75)	5.841	0.054
Somatic Problems	1.07 (1.28)	1.20 (1.55)	1.19 (1.41)	0.123	0.940
Attention Deficit/Hyperactivity Problems	2.34 (2.47)	2.60 (2.27)	2.03 (1.86)	0.426	0.808
Oppositional Defiant Problems	2.07 (1.60)	1.80 (1.23)	1.56 (1.18)	1.405	0.495
Conduct Problems	1.00 (1.56)	0.90 (1.10)	2.53 (2.52)	10.421	0.005 *
Internalizing Problems	7.97 (6.50)	8.50 (5.25)	5.44 (4.38)	4.246	0.120
Externalizing Problems	5.03 (4.34)	3.50 (2.68)	5.75 (4.64)	1.660	0.436
Total	22.65 (16.77)	23.60 (8.42)	21.94 (14.51)	0.629	0.730

DASS-21: Depression Anxiety and Stress Scale; SDQ: Strengths and Difficulties Questionnaire; CBCL: Child Behaviour Checklist for Children; * *p* < 0.05.

## Data Availability

Raw data will be uploaded on www.Zenodo.org immediately after the acceptance of the manuscript and they will be available upon reasonable request to the authors A.G.U and A.S.

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
