# Peer review of "Emotional and Behavioural Adjustment in Children and Adolescents with Short Stature vs. Their Normal-Statured Peers"

_jcm, 2025, doi:10.3390/jcm14020538_

Round 1

Reviewer 1 Report

Comments and Suggestions for Authors

I read the text carefully and found the following:

1. I wonder if the title could be improved. This is not mandatory, but consider the appropriateness of a title such as "Psychological adjustment in young people with short stature compared to their peers of average height."

2. In the abstract section, you have stated up front what the abbreviation GHD stands for, but it is not clear what the abbreviation SDS stands for (line 28). This makes the abstract text hard to follow.

3. Phrasing throughout the entire manuscript is improper when talking about scores on a particular scale or subscale. It is improper to say "higher levels of withdrawn/depressed subscale" (line 35), but the proper expression is "higher levels of withdrawn/depressed subscale score". On line 229, you say "higher levels of the withdrawn/depressed subscale of CBCL". You need to correct all similar instances in the manuscript.

4. You need to include some statistics showing that the three groups are equivalent in terms of other variables that might influence the observed differences. For example, on line 184, you state that 'Age, gender composition and school attendance were similar in the three subgroups', but you do not explain why. A Chi-Square statistic will show whether there is a significant difference between the observed and expected frequency of male and female subjects respectively across the three independent groups. You have also presented the mean age for each group, but you have not shown by a statistic whether the difference between the three groups is significant. There is a similar problem with the school attendance variable, which depends on how it is measured. We need to show that the three groups are equivalent with respect to these variables.

5. One of the most problematic aspects of the research is that you do not report at all how the psychometric scales behave, even though we understand that the sample sizes are small. We must report a reliability indicator, like Alpha-Cronbach's or Split-Half, to show that the items in the group fit together. It's not enough to know that these Italian language versions have been validated; we also need to check the internal consistency in each research study.

6. Including the effect size for each significant statistic you find would show professionalism. Try G*Power Statistics downloadable at G*Power .This would also improve the 'Discussion' section, as you can relate your conclusions to larger or smaller effects.

Author Response

Reviewer 1

We acknowledge the Reviewer for carefully evaluating our manuscript, which is aimed at improving the quality of our work further.

R1. I wonder if the title could be improved. This is not mandatory, but consider the appropriateness of a title such as "Psychological adjustment in young people with short stature compared to their peers of average height."

A1. Thank you for your comment. However, we would like to maintain the original title since we believe it is more appealing and informative regarding the contents of the paper.

R2. In the abstract section, you have stated up front what the abbreviation GHD stands for, but it is not clear what the abbreviation SDS stands for (line 28). This makes the abstract text hard to follow.

A2. We have stated that SDS means standard deviation score.

R3. Phrasing throughout the entire manuscript is improper when talking about scores on a particular scale or subscale. It is improper to say "higher levels of withdrawn/depressed subscale" (line 35), but the proper expression is "higher levels of withdrawn/depressed subscale score". On line 229, you say "higher levels of the withdrawn/depressed subscale of CBCL". You need to correct all similar instances in the manuscript.

A3. Thank you for your valuable comment. We have corrected the sentences in the entire manuscript.

R4. You need to include some statistics showing that the three groups are equivalent in terms of other variables that might influence the observed differences. For example, on line 184, you state that 'Age, gender composition and school attendance were similar in the three subgroups', but you do not explain why. A Chi-Square statistic will show whether there is a significant difference between the observed and expected frequency of male and female subjects respectively across the three independent groups. You have also presented the mean age for each group, but you have not shown by a statistic whether the difference between the three groups is significant. There is a similar problem with the school attendance variable, which depends on how it is measured. We need to show that the three groups are equivalent with respect to these variables.

A4. We have performed a Fisher test to show differences between the frequencies of males and females and school attendance in the three subgroups. We have also performed a one-way ANOVA to show differences in means of age between the three subgroups. These results are now presented in line  196.

R5. One of the most problematic aspects of the research is that you do not report at all how the psychometric scales behave, even though we understand that the sample sizes are small. We must report a reliability indicator, like Alpha-Cronbach's or Split-Half, to show that the items in the group fit together. It's not enough to know that these Italian language versions have been validated; we also need to check the internal consistency in each research study.

A5. Since the sample was too small to perform a valid Alpha’s Cronbach, we have performed correlations among items of the same subscale for each questionnaire in order to provide a measure of internal consistency. This approach serves as a valid alternative to calculating Cronbach's alpha, particularly when the sample size is small. These results are now reported in lines 144-148; 161-163; 171-173. The low internal consistency has been added as a limitation of the study.

R6. Including the effect size for each significant statistic you find would show professionalism. Try G*Power Statistics downloadable at G*Power. This would also improve the 'Discussion' section, as you can relate your conclusions to larger or smaller effects.

A6. As requested, ε² effects sizes have been included in the statistical analysis and Discussion. Please see lines 241;244;245, respectively.

Reviewer 2 Report

Comments and Suggestions for Authors

Thank you for the opportunity to review this manuscript submission. The manuscript is incredibly clear, well-structured, and understandable for someone without specific experience working with youth of short stature.

Brief summary: The aim of this manuscript is to compare the emotional and behavioral conditions of youth with not-GH dependent short stature, idiopathic GHD, and normal height through the use of a cross-sectional research study. This study contributes data that youth with idiopathic GHD exhibited higher levels of the withdrawn/depressed subscale of CBCL compared to their peers with normal height, while no significant differences were observed between subjects with not-GH dependent short stature and those with normal stature.

General concept comments:
A study aim is noted, but no explicit hypothesis is stated.
Therefore, the experimental design is unable to be fully evaluated as appropriate for testing the hypothesis.

Specific comments: Line 28: Is “subjects” the best term to use? Subject may inadvertently contradict the principles of informed consent and participant autonomy. “Participant” may better acknowledges the active role of individuals. Please consider adding a unit of measurement each time “mean height SDS: -2.10 +0.57” is referenced for international readers and consistency with mean age wording.

Line 30: Should the mean age and standard deviation be noted as 11.3 ± 1.93 years?

Line 92: Missing closed parenthesis? “)”

Line 94: Was the observations of short stature in other family members limited to family of origin or extended family as well?

Line 114-116: What is meant by in-person method - Interview?

Line 146: The Child Behavior Checklist for Children also includes the Teacher’s Report Form and Youth Self-Report profiles. Were these measures used or included in the results? Line 212 phrasing reads as youth self-report.

Line 174-175: Assuming both Bonferroni and Holm-Bonferroni methods are being used to control family-wise error rate, is using both methods together necessary?

Line 228-231: This phrasing is a bit confusing. I think you mean, “there are no significant differences were observed between subjects with not-GH dependent short stature and those with normal stature.” Without the clarification of not-GH dependent short stature, it reads as contradictory, as if subjects with idiopathic GHD and subjects with short stature are synonymous.

Tables are appropriate, properly show the data, are easy to interpret and understand. Table 1: How were demographic data collected - Survey? The Group 1 School Attendance data (Primary school 41%, Secondary 58%) only adds to 99%. Is this a rounding error?

The manuscript is scientifically sound with the exception of a missing hypothesis. Results are reproducible based on the details given in the methods section. The data are interpreted appropriately and consistently throughout the manuscript. The conclusions are consistent with the evidence and arguments presented. Institutional Review Board, Informed Consent, and Data Availability statements are all adequate. Limitations of generalizability and delimitation efforts are noted. References are mostly more than 5 years old. However, this may be reflective of using original sources as opposed to outdated trends, methodologies or findings.

Author Response

Reviewer 2

Thank you for the opportunity to review this manuscript submission. The manuscript is incredibly clear, well-structured, and understandable for someone without specific experience working with youth of short stature.

We acknowledge the Reviewer for carefully evaluating our manuscript, which is aimed at improving the quality of our work further.

R1. A study aim is noted, but no explicit hypothesis is stated. Therefore, the experimental design is unable to be fully evaluated as appropriate for testing the hypothesis.

A1. We agree that an experimental design is supposed to state one or more hypotheses to be tested. However, as repeatedly mentioned in the aim of the study (abstract: line 23-26, introduction: line 81-83, discussion: line 237-239), this work is a cross-sectional study aimed at exploring the prevalence of certain characteristics without establishing a priori hypotheses of causal relationships between variables.

R2. Is “subjects” the best term to use? “Subject” may inadvertently contradict the principles of informed consent and participant autonomy. “Participant” may better acknowledges the active role of individuals. Please consider adding a unit of measurement each time “mean height SDS: -2.10 +0.57” is referenced for international readers and consistency with mean age wording.

A2. As requested we have changed “subjects” with “participants” throughout the entire manuscript.

As far as the reviewer's comment regarding the need to add a unit to the standard deviation score (SDS) is concerned, SDS is a score which does not obviously require the unit of measurement, as reported in other recent papers using the same parameter:

  • Ogawa, T., Narusawa, H., Nagasaki, K., Kosaki, R., Naiki, Y., Aramaki, M., ... & Kagami, M. (2024). Temple Syndrome: Comprehensive Clinical Study in Genetically Confirmed 60 Japanese Patients. The Journal of Clinical Endocrinology & Metabolism, dgae883.
  • Dumont, E., Tiemens, D. K., Draaisma, J. M., Kleimeier, L. E., van Druten, D., & Mulkens, S. (2024). Do children with a Noonan syndrome-like RASopathy and avoidant/restrictive food intake disorder benefit from behavioral therapy?. European Journal of Pediatrics184(1), 100.
  • Alashwal, A. A., Al-Fattani, A., Ramzan, K., Imtiaz, F., & Binladen, A. (2024). Long-term treatment for Laron syndrome with IGF-1 injection over 22 years in Saudi: A cohort study. Hormone Research in Paediatrics, 1-23.)

R3. Line 30: Should the mean age and standard deviation be noted as 11.3 ± 1.93 years?

A3. Thank you. We have checked and confirmed.

R4.Line 92: Missing closed parenthesis? “)”

A4. Thank you, we have checked and corrected.

R5.Line 94: Was the observations of short stature in other family members limited to family of origin or extended family as well?

A5. Short stature was familiar (i.e. family of origin). Short stature was predominantly present in one (or two) of the parents; in a few cases, short stature was present in one (or more) brother and/or sister.

R6.Line 114-116: What is meant by in-person method - Interview?

A6. We have deleted this probably confusing term.

R7. Line 146: The Child Behavior Checklist for Children also includes the Teacher’s Report Form and Youth Self-Report profiles. Were these measures used or included in the results? Line 212 phrasing reads as youth self-report.

A7. We have used the CBCL completed by the caregivers only.

R8. Line 174-175: Assuming both Bonferroni and Holm-Bonferroni methods are being used to control family-wise error rate, is using both methods together necessary?

A8. Thank you. We used the Holm-Bonferroni method (lines 185). Holm-Bonferroni is considered the better option because it controls Type I error adequately, being more powerful than the traditional Bonferroni correction.

R9. Line 228-231: This phrasing is a bit confusing. I think you mean, “there are no significant differences were observed between subjects with not-GH dependent short stature and those with normal stature.” Without the clarification of not-GH dependent short stature, it reads as contradictory, as if subjects with idiopathic GHD and subjects with short stature are synonymous.

A9. Not-GH-dependent short stature includes individuals with FSS and CGD (group 1). In order to avoid confusion, we have modified the sentence (line 225)

R10. Tables are appropriate, properly show the data, are easy to interpret and understand. Table 1: How were demographic data collected - Survey?

A.10 Thank you. Demographic data were self-reported by participants before completing the questionnaires.

R11.The Group 1 School Attendance data (Primary school 41%, Secondary 58%) only adds to 99%. Is this a rounding error?

A11. Sorry for the mistake. We have checked and corrected the table.

R12.The manuscript is scientifically sound with the exception of a missing hypothesis. Results are reproducible based on the details given in the methods section. The data are interpreted appropriately and consistently throughout the manuscript. The conclusions are consistent with the evidence and arguments presented. Institutional Review Board, Informed Consent, and Data Availability statements are all adequate. Limitations of generalizability and delimitation efforts are noted. References are mostly more than 5 years old. However, this may be reflective of using original sources as opposed to outdated trends, methodologies or findings.

A12. Thank you for the appreciation of our work.

Round 2

Reviewer 1 Report

Comments and Suggestions for Authors

I am, overall, satisfied with the changes and do not request any further improvements.